

# The G4Foam Experiment:
## Global Climate Impacts of Regional Ocean Albedo Modification

Corey J. Gabriel*, Alan Robock, Lili Xia, and Brian Zambri

Department of Environmental Sciences, Rutgers University, New Brunswick, NJ, USA

Submitted to *Atmospheric Chemistry and Physics*
Special Issue: The Geoengineering Model Intercomparison Project

September, 2016

*To whom correspondence should be addressed: Corey J. Gabriel, Department of Environmental Sciences, Rutgers University. 14 College Farm Road. New Brunswick, NJ 08901-8551. E-mail: corey@envsci.rutgers.edu.





**Abstract.**  Reducing insolation has been proposed as a geoengineering response to global
warming.  Here we present the results of climate model simulations of a unique Geoengineering
Model Intercomparison Project Testbed experiment to investigate the benefits and risks of a
scheme that would brighten certain oceanic regions.  The National Center for Atmospheric
Research CESM-CAM4-CHEM global climate model was modified to simulate a scheme in
which the albedo of the ocean surface is raised over the subtropical ocean gyres in the Southern
Hemisphere.  Like the commonly studied stratospheric geoengineering and marine cloud
brightening proposals, this ocean albedo modification scheme is not currently possible.
However, a stable, nondispersive foam, comprised of tiny, highly reflective microbubbles has
been developed under idealized conditions, and, hence, a geoengineering scheme which
simulates the effects of large-scale deployment of these microbubbles is appropriate to study at
this time.  One goal of this scheme is to cool Earth without reducing monsoon rainfall.  We
conducted three ensemble members of a simulation (G4Foam) from 2020 through 2069 in which
the albedo of the ocean surface is raised to 0.15 over the three subtropical ocean gyres in the
Southern Hemisphere, at the same time as increasing the radiative forcing with the RCP6.0
(representative concentration pathway resulting in +6 W m$^{-2}$ radiative forcing by 2100) scenario,
and then continuing the simulation for 20 more years with RCP6.0.  Global mean surface
temperature in G4Foam is 0.6 K lower than RCP6.0, with statistically significant cooling relative
to RCP6.0 south of 30°N and an increase in rainfall over land, most pronouncedly in the tropics
during the June-July-August season, relative to both G4SSA (specified stratospheric aerosols)
and RCP6.0.  Heavily populated and highly cultivated regions throughout the tropics, including
the Sahel, Southern Asia, the Maritime Continent, Central America and much of the Amazon
experience a statistically significant increase in precipitation minus evaporation.  The
temperature response to the relatively modest global average forcing of –1.5 W m$^{-2}$ is amplified
through a series of positive cloud feedbacks, in which more shortwave radiation is reflected.  The
precipitation response is primarily the result of the intensification of the southern Hadley cell, as
its mean position migrates northward and away from the Equator in response to the asymmetric
cooling.





## 1 Introduction
### 1.1 Background

The current rate of increase in global mean surface temperature is unprecedented in the
last 1,000 years (Marcott et al., 2013). This warming is co-occurring with a rate of increase in
the atmospheric concentration of carbon dioxide ($CO_2$) and other greenhouse gases that exceeds
any other increase in the recent record by one or two orders of magnitude, and the atmospheric
concentration of $CO_2$ is higher now than at any time in the last 650,000 years (Siegenthaler et al.,
2005). It is extremely likely that the warming since 1950 is primarily the result of anthropogenic
emission of heat-trapping gases rather than natural climate variability (IPCC, 2013).
A global consensus on the need for mitigation has emerged as the December 2015 Paris
Agreement at the $21^{st}$ Conference of the Parties of the United Nations Framework Convention on
Climate Change (UNFCCC). However, the UNFCCC lacks enforcement mechanisms, unlike the
successful 1987 Montreal Protocol on Substances That Deplete the Ozone Layer, which legally
obliged the signatories to phase out chlorofluorocarbons by 1998. Further, the Paris Agreement
relies on nonbinding pledges from nations and avoids explicit emissions targets for specific
countries. Our ability to reach the aspirations set out in the Paris Agreement will largely rely on
these voluntary contributions to mitigation efforts. The nature of these voluntarily contributes
are to be determined on the national level. Even then, attaining the Paris targets will likely
require large-scale removal of carbon dioxide from the atmosphere (Fuss et al. 2014; Sanderson
et al. 2016), which would require carbon dioxide removal to work as intended.
Therefore, even post-Paris, most realistic emission scenarios still send the global mean
surface temperature to more than 2 K above preindustrial. Sophisticated adaptation efforts,
including the planning of infrastructure projects in anticipation of an increase in extreme events,
are underway (Miller et al., 2013; Fischbach et al., 2016). However, resources are finite and the
planning of and execution of sophisticated large-scale projects necessary for adaptation has been
largely limited to the world's richest countries.
Under these conditions, it is easy to see that the combination of mitigation and adaptation
may be insufficient to spare us from progressively more disruptive impacts of global warming.
This sad reality elevates the discussion of solar radiation management (SRM) geoengineering to
relevance. SRM has been proposed as a method of reducing global mean temperature, thereby
ameliorating many of the negative effects of global warming (Crutzen, 2006). The most
discussed SRM approach involves injection of sulfur dioxide ($SO_2$) into the tropical stratosphere.
The $SO_2$ reacts with water and aqueous sulfuric acid ($H_2SO_4(aq)$) is formed. It is theorized that
this will then form a relatively stable layer of $H_2SO_4(aq)$ droplets with a diameters of less than
about 1 μm and mix globally through the Brewer-Dobson circulation, reflecting incoming
shortwave (SW) radiation at all latitudes. Other suggested SRM geoengineering methods
include marine cloud brightening (Jones et al., 2009; Rasch et al., 2009; Latham et al., 2010) and
surface albedo modification (Irvine et al., 2010; Cvijanovic et al., 2015). Each of these methods
has the potential to cool Earth's surface, but each comes with known potential side effects, many
of which would be undesirable. For example, Robock (2008, 2014) enumerates and describes
specific risks and benefits of stratospheric geoengineering.
Here we present a Geoengineering Model intercomparison Project (GeoMIP) testbed
experiment (Kravitz et al., 2011, 2016), consisting of the novel implementation of an ocean
surface albedo modification scheme in a climate model, which simulates the placement of a
reflective foam, consisting of microbubbles, on the ocean surface. The albedo of the ocean
surface is raised from a daily average 0.06 to a fixed daytime value of 0.15 over the subtropical

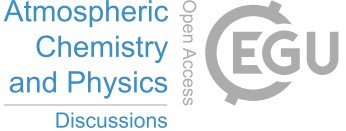

ocean gyres in the Southern Hemisphere, specifically 20°N-20°S, 90°W-170°W (South Pacific),
20°N-20°S, 30°W-0°E (South Atlantic) and 20°N-20°S, 55°E-105°E (South Indian) (Fig. 1).
This is the G4Foam experiment, which simulates a particular implementation of an idealized
form of the technology described by Aziz et al. (2014), where stable, reflective foam suitable for
use as SRM in ocean regions with limited nutrients that support little marine life is made in the
laboratory.
The broad idea of microbubble deployment as a form of SRM is explored by Seitz
(2010). Here we only examine the potential benefits and risks of such a scheme, and do not
advocate deployment of any form of geoengineering even if it were presently possible. Robock
(2011) has cautioned against the potential implications of ocean albedo modification as presented
by Seitz (2010).
Stratospheric aerosol injections (SAI) are the most discussed, and given current state of
research the most feasible, form of geoengineering (Dykema et al., 2014, Keith et al., 2014).
Implementation of the G4Foam regional ocean albedo modification scheme could be considered
with or without concurrent SAI. G4Foam could be used as a potential SSI concurrent scheme
aimed at correcting possible negative impacts on the hydrological cycle brought about by
ongoing SAI. G4Foam is also a potential alternative to SAI with a far different latitudinal
distribution of benefits. The focus here is solely on the second scenario, as it allows for the
elucidation of the impacts of the G4Foam experiment forcing alone.
While G4Foam and SAI are aimed in part at reducing surface temperature, their
objectives vary in two profound ways. First, G4Foam primarily aims to enhance water supply by
repositioning the tropical rain belts into highly populated, heavily cultivated areas. Second,
unlike SSI, where the high latitude Northern Hemisphere (NH) cools preferentially, G4Foam
preferentially cools Earth south of 30°N.

**1.2 Motivation and Research Question**

The goal of climate engineering is to reduce global warming without introducing
additional risks. One of the major issues with stratospheric aerosols is a reduction of
precipitation, especially for summer monsoons. The intervention discussed here attempts to
reduce warming without reducing precipitation.
Is it possible to cool the planet while concurrently maintaining or increasing precipitation
in highly populated and heavily cultivated regions, particularly in regions dependent on monsoon
precipitation? We must begin by determining whether a forcing can be applied in a global
climate model (GCM) that will result in the model responding with a northward and landward
shift of tropical precipitation needed to achieve our objective. To that end we conducted
simulations with The Community Earth System Model 1/Community Atmospheric Model 4 fully
coupled to tropospheric and stratospheric chemistry (CESM1 CAM4–Chem) model (Lamarque
et al., 2012; Tilmes et al., 2015, 2016). We ran the model with horizontal resolution of 0.9° x
1.25° lat-lon and 26 levels from the surface to about 40 km (3.5 mb), as was done for G4SSA
(specified stratospheric aerosol) by Xia et al. (2016).
The experiments consisted of three ensemble members of a simulation from 2020-2089 in
which the ocean surface albedo is raised as described above from a daytime average of 0.06 to a
daytime constant 0.15 on the SH subtropical ocean gyres for 50 years, 2020-2069, and then
returned to unforced values from 2070-2089 to assess termination. Our hypothesis is that the
tropical rain belts will move northward largely as a result of increased moisture convergence
over land regions, particularly during Northern Hemisphere (NH) summer (June-July-August,
JJA) in NH monsoon regions. Enhanced divergence over the already strong subtropical highs,



due to increased subsidence over the increased albedo ocean regions in the subtropical Southern
Hemisphere (SH), would help the cooler air from the forced subtropical regions advect
throughout the SH troposphere.  The asymmetric cooling would force changes in the Hadley
Cell, enhancing cross equatorial flow, which would cool the surface in the NH tropics, especially
during JJA, when heat related mortality and morbidity is highest.  Finally, the resulting cooling
of low latitude NH land areas would not dampen the monsoon.  The wet season monsoon
circulation is initiated and maintained by the moist static energy gradient, not the surface
temperature gradient.  A wetter, more cloudy land mass will strengthen, not dampen the
circulation relative to a warmer, drier continent (Hurley and Boos, 2014), especially with a
cooler, lower specific humidity environment under the descending branch of the meridional
circulation.
The strength of this response will be very sensitive to the cloud feedbacks that result from
the surface albedo forcing.  The basis of this comprehensive hypothesis is described in detail,
below, specifically in sections 1.3 and 1.4.  The details of the experiment are discussed in detail
in section 2.

**1.3  Stratospheric geoengineering weakens the hydrological cycle**

With global warming, low-level specific humidity will increase by about 7% $K^{-1}$ within
the tropical planetary boundary layer.  This response will be spatially homogeneous throughout
the tropics.  However, the precipitation response will be different.  Increased moisture
convergence in areas that already get a lot of precipitation will result in the "wet getting wetter,"
while increased moisture divergence in dry areas will result in the "dry getting drier" (Held and
Soden, 2006).
The "rich get richer, poor get poorer" paradigm does not hold up in an SRM world, where
the response is very different from that under global warming.  Based on the results of an
observational study, Trenberth and Dai (2007) pointed out the possibility that drought,
particularly in the tropics, could result from geoengineering.  This is based on a study of tropical
volcanism, which provides a good analog to stratospheric SRM, particularly SSI approaches.
They found that after the eruption of Mount Pinatubo (15°N) in 1991, precipitation dropped over
land and record drops in runoff and river discharge were observed from October 1991 to
September 1992.
Using a comprehensive atmosphere-ocean general circulation model, the National
Aeronautics and Space Administration Goddard Institute for Space Studies ModelE, Robock et
al. (2008) documented a reduction in rainfall over China and India, especially during NH
summer, which is associated with a weakened monsoon.  Changes in precipitation patterns also
imply a southward shift of the intertropical convergence zone (ITCZ) over the Atlantic and
Pacific Oceans.  Additional similar experiments showed similar results.
In the GeoMIP G1 experiment, abrupt 4xCO2 forcing is applied to a year 1850 control
climate, and the $CO_2$ forcing is fully offset by concurrent solar dimming (Kravitz et al., 2011).
To explain the different hydrological cycle responses to surface and atmosphere energy balance
responses to greenhouse gas and solar forcings, Kravitz et al. (2013) compared G1 with the
experiment abrupt 4xCO2 and the Pre-Industrial Control experiment (picontrol), which is the
reference experiment for both abrupt 4xCO2 and G1.  They found a reduction in precipitation in
G1 relative to abrupt 4xCO2 that can be separated into a fast component, due to the radiative
response, which scales with the applied forcing, and a slow response, which is a temperature
effect.  Tilmes et al. (2013) analyzed the hydrological cycle in most of the GeoMIP participating
Coupled Model Intercomparison Project 5 (CMIP5) (Taylor et al., 2012) models by comparing



abrupt 4xCO2, picontrol, and G1. They found a robust reduction in global monsoon rainfall,
including in the Asian and West African monsoon regions in G1 relative to both abrupt 4xCO2
and picontrol. Haywood et al. (2013) explored the impact of SSI in one hemisphere only and
found a movement of the ITCZ away from the hemisphere that was cooler as a result of the
asymmetric SSI.
This consensus about the potential for reduced tropical rainfall under a regime of
stratospheric SRM motivates us to identify an alternative or SSI-adjunctive geoengineering
approach that could cool the planet, without reducing monsoon precipitation in highly cultivated
areas.

## 1.4 Extratropical forcing impacts the position of the ITCZ

Under global warming tropical rainbelts will move toward the hemisphere that warms
more (Chiang and Bitz, 2005, Frierson and Hwang, 2012). Early atmosphere-ocean coupled
models involved prescribed clouds. Increasing low cloud cover, and thereby inducing cooling, in
one hemisphere relative to the other caused the tropical rainbelts over the Pacific Ocean to move
toward the other hemisphere (Manabe and Stouffer, 1980). The impacts of asymmetric heating
of the hemispheres became highly relevant during the Sahel drought. Much of the rainfall deficit
during the devastating 20-30 year drought can be attributed to cooling initiated by increased
tropospheric sulfate emissions in the NH (Hwang et al., 2013). The forced cooling over the NH
was enhanced by a positive dynamical feedback in the North Atlantic Ocean. This hemispheric
asymmetry moved Earth's energy flux equator, the vertical surface boundary where the vertically
integrated meridional energy flux is zero, southward (Broccoli et al. 2006; Kang et al. 2008).
Hence, the ITCZ and associated tropical rainbelts migrated south. Since the Sahel is at the
northern margin of the ITCZs annual migration, or at the northern terminus of the West African
monsoon, southward displacement of the ITCZ led to a devastating drought (Folland, 1986).
Broccoli et al. (2006) diagnosed the energy balance mechanism that causes the ITCZ to
shift in response to asymmetric heating of the extratropics. Using models of varying complexity,
Broccoli et al. (2006) imposed an anomalous cooling of the NH, either via a last glacial
maximum simulation, or via hosing of the North Atlantic. The heating asymmetry causes the
extratropics in the NH to demand more heat and the extratropics in the SH to demand less heat.
Since cross equatorial heat transport is achieved principally via the Hadley Cell, the SH Hadley
Cell strengthens, particularly in austral summer, in response to the NH cooling, and net energy
flow in the upper branch intensifies, redistributing energy into the NH from the relatively warm
SH.
Net flow of energy in the Hadley cell is achieved by the flow of moist static energy,
which flows in the direction of the upper troposphere branch of the Hadley Cell. This is because
moist static energy is higher at higher altitudes in the troposphere due to the increased
contribution of the geopotential energy term overwhelming the moisture and internal energy
terms in the moist static energy equation for the high altitude air. Net transport of energy,
occurring in the upper branch of the Hadley cell from the SH to the NH, leads to increased
moisture advection to the SH in the lower branch of the Hadley Cell. This redistribution of
energy causes the ascending branch of the Hadley cell to migrate to the warmer SH where
moisture convergence is increased and convective quasi-equilibrium is achieved under the
relatively narrow poleward shifted ascending branch of the stronger SH winter Hadley Cell.
This mechanism leads to the southward-displaced tropical rain belts (Broccoli et al., 2006).
This result is consistent with Lindzen and Hou (1988), who used a relatively simple
model to show that even a small movement of maximum heating poleward into one hemisphere





causes great asymmetry in the Hadley Cell, with the winter cell intensifying tremendously and
the summer cell becoming rather modest.  Further simplified model studies continued to
elucidate the mechanism of extratropical forcing of the ITCZ.  Kang et al. (2008) heated the SH
poleward of 40°S while cooling the NH poleward of 40°N by an equal amount.  The overall
energy in the system was unchanged.  The atmosphere was coupled to a simple swamp ocean.
Interhemispheric heat exchange could only be achieved through the atmosphere.  In this model,
the energy flux equator and tropical rain bands moved to the SH.
In the real world, the ocean also plays a vital role in pushing the ITCZ into the warmer
hemisphere.  This is well-described by Xie and Philander (1994).  They used a mixed layer ocean
model, coupled with a simple atmosphere, which was capable of transporting momentum and
heat to the ocean.  This impacts SST via surface wind-induced mixing.  Asymmetric heating
produced a single ITCZ that forms in one hemisphere only.  The atmosphere forced enhanced
equatorial upwelling in the asymmetric Hadley Cell solution.  This pushed the ITCZ toward
warmer SSTs in the warmer hemisphere.
GCM results confirm this mechanism and connect the changes due to northward
displacement of the ITCZ with the onset of active periods in the Asian summer monsoon (Chao
and Chen 2001).  It is evident that a geoengineering technique that could preferentially cool the
SH could shift the tropical rain bands northward.  However, in a GCM there are clouds.  How
would clouds respond in the hemisphere cooled by geoengineering?  Would clouds change in the
area being directly cooled?  Would a cooling of the subtropics either directly, or indirectly via
eddy flux from the artificially cool high latitudes, cause an increase in subtropical subsidence?
Would this increase in the sinking of air above the intensified subtropical highs cause water
vapor to be trapped in the lower troposphere, forming low clouds and suppressing water vapor
mixing into the free troposphere, where the water vapor may instead be used up in formation of
high clouds, which tend to reduce outgoing longwave radiation?  Informed by these established
diagnostic mechanisms associated with the impacts of asymmetric heating of the hemispheres,
we seek to concurrently cool the entire SH and the NH tropics, modestly cool the NH
extratropics and, most importantly, induce an anomalous overturning circulation and redistribute
rainfall from ocean to land and from south to north across the tropics.
**2. Methods**
**2.1 Design of experiment and model configuration**
Figure 1 shows the regions selected for albedo enhancement.  These regions were chosen
because of their low cloud fraction, low wind speeds, weak currents, and lack of biological
productivity in addition to the likelihood that the surface cooling in this area will advect well
within the SH.
We used the Community Land Model (CLM) version 4.0 with prescribed satellite
phenology (CLM4SP) instead of the version of CLM with a carbon–nitrogen cycle, coupled with
CAM4–chem.  Vegetation photosynthesis is calculated under the assumption of prescribed
phenology and no explicit nutrient limitations (Bonan et al., 2011, Xia et al., 2016). Dynamic
vegetation is not turned on in this study.  The ocean model does not include any biogeochemical
responses.
The fundamental question we wish to answer concerns representation of the physical
processes that lead to realistic simulation tropical precipitation.  The Asian monsoon is of great
importance in that investigation.  Fortunately, monsoon processes and regimes are depicted well
in our atmospheric component, CAM4 (Meehl et al., 2012).  Some important features of CAM4
that illustrate its very good monsoon representation include the amount and location of





precipitation over the southern Tibetan Plateau and over the Western Ghats (a mountain range
near the west coast of south India). This is improved when compared to earlier versions of the
model. The rain shadow leeward of this range is often not resolved by GCMs, however CAM4
shows some evidence of this rain shadow. These changes related to orography and horizontal
resolution are important and likely generalize to similar land surface features outside of India,
where model biases have not been as carefully studied as they have been in heavily populated
southern India. This improvement can be attributed to the CCSM4 finite-volume dynamical
core, which replaces the spectral version of the CCSM3 and the interconnected higher horizontal
resolution. Additionally, large-scale features are improved. For example, the representation of
the ITCZ during NH winter southward migration over the maritime continent is improved
(Meehl et al., 2011).
There is an important process associated with monsoon precipitation, however, that is
pervasively wrong across CMIP5. Zonal mean absorbed shortwave radiation is too high over the
southern ocean (Kay et al., 2016). This cloud problem leads to a warmer Southern Ocean, which
leads to anomalous SH atmospheric eddy flux to the subtropics from the extratropics, potentially
damping the cooling response of our negative surface radiative forcing in the subtropical oceans.
The effect of a transfer of heat from the SH extratropics into the Hadley Cell already causes a
relatively weak negative bias in the amount of interhemispheric heat transport from the south to
north. Therefore, the manifestation of this bias in G4Foam would be to partially offset our
imposed cooling, lessening the need for interhemispheric energy transport to the SH and
suppressing the surface return flow of moisture advection into the NH. Lower than observed
interhemispheric energy transport would be associated with a weaker Asian monsoon. However,
this feature is equally present in our G4Foam experiment and its control experiments so will not
affect the differences.
We compare G4Foam to two control experiments. First is a specific sulfate injection
scenario, G4 Specified Stratospheric Aerosol (G4SSA; Xia et al., 2016). They used a prescribed
stratospheric aerosol distribution roughly analogous to annual tropical emission into the
stratosphere (at 60 mb) of 8 Tg $SO_2$ $yr^{-1}$ from 2020 to 2070. This produces a radiative forcing of
about $-2.5$ W m$^{-2}$. The G4SSA forcing ramps down from 2069-2071 and then continues without
additional forcing from 2072-2089. In G4SSA tropospheric aerosols are not affected by the
prescribed stratospheric aerosols. Therefore we cannot evaluate how stratospheric aerosols
would actually fall out and impact the chemistry, dynamics and thermodynamics of the
troposphere from this experiment. Neely et al. (2015) offers more detail on the prescription of
stratospheric aerosols in CAM4–Chem. The second control, which serves as the reference
simulation, for both G4Foam and G4SSA is the Representative Concentration Pathway 6.0
(RCP6.0) (Meinshausen et al., 2011) from 2004 to 2089. We have run three ensemble members
each for G4Foam, G4SSA, and RCP6.0.

## 2.2 Ocean albedo enhancement approach

The advent of a plausible technology to make quantities of long lasting foam, or
engineered microbubbles to enhance ocean albedo, elevates ocean albedo modification to a status
of plausibility that is roughly equivalent to that of other widely discussed potential
geoengineering methods. Seitz (2010) speculated that since air-water and air-sea interfaces are
similarly refractive, dispersing microbubbles onto the surface of the ocean would reflect sunlight
in much the same way as cloud droplets do. While engineering refractive or stable foams is
commonly done and applied in both food science and firefighting, engineering a stable and
refractive foam appropriate for a geoengineering scheme appeared fanciful until Aziz et al.





(2014) produced a long lasting refractive foam made with biodegradable and non-toxic additives.
Aziz et al. identified foam lifetime of three months or more per microbubble as lasting long
enough that the input of energy to create the microbubbles would not be prohibitive.  After
experimenting with protein-only solutions, Aziz et al. (2014) added "high methyl ester pectin to
type A gelatin" and created a foam in salt water, which was still intact and stable at the cessation
of the experiment after 3 months.  The reflectance of the foam was about 50%, which is
comparable to that of whitecaps.  The creation of these stable microbubbles makes enhancing
ocean albedo in this manner "feasible" (Aziz et al. 2014).
Safe, stable, highly reflective microbubbles were created in saltwater in a laboratory.
However, this technology – like other SRM techniques – is still a long way from deployment.
For example, what effect would bacteria in the ocean have on the surfactant?  There are a
number of other potential risks associated with microbubble deployment, even if the feasibility
issues are set aside.  Robock (2011) pointed out that vertical mixing in the ocean, changes in
ocean circulation, impacts on photosynthesis, and risks to the biosphere could all impair the
efficacy of this geoengineering approach.  Robock (2011) also pointed out that a cooler ocean
would serve as a more effective $CO_2$ sink, helping to offset the $CO_2$ increase that comes about as
a feedback of warming.  Other potentially attractive attributes of this technique include the
possibility that it could be deployed exclusively in the 20% of the world's oceans that are not
biologically active (Aziz et al. 2014) and therefore have little impact on the biosphere, that there
would be no risk to ozone in the stratosphere.

**3 Results**

The following results compare the G4Foam climate with the climates in G4SSA and
RCP6.0 averaged over the period 2030-2069.  While G4Foam and G4SSA forcing commences in
2020, the first ten years of both experiments are a period of transition.  For that reason 2020-
2029 is discarded from our comparisons.

**3.1  Temperature and cloud response**

The primary purpose of G4Foam is to reduce global mean surface temperature without
reducing monsoon precipitation. G4Foam would reduce global mean surface temperature
relative to RCP6.0 by 0.60 K and global mean land surface temperature by 0.51 K relative to
RCP6.0.  In JJA, G4Foam is 0.70 K cooler than RCP6.0 over land in the tropics, 20°S-20°N.
(Table 1).  This JJA cooling in the tropics is of particular importance due to the dense population
and heavy agricultural demand in the tropics, particularly north of the equator.  The G4Foam
cooling is achieved by a change in net clear sky top of atmosphere flux in G4Foam of $-1.5$ W m$^{-2}$
(Figure 2).
G4Foam does not achieve the same amount of cooling as G4SSA, which would reduce
global mean surface temperature by 0.92 K.  This is achieved by a change in net clear sky top of
atmosphere shortwave flux in G4Foam of $-4.0$ W m$^{-2}$.  On a global mean basis and relative to
RCP6.0, G4Foam achieves 66% of the cooling produced by G4SSA, while applying only 38% as
much forcing as G4SSA (Figure 2).  This implies that much of the G4Foam cooling is the result
of positive feedbacks, which enhance the cooling, as discussed below.
Figure 3 shows a comparison of the spatial distribution of surface temperature changes
between G4Foam and G4SSA and between G4Foam and RCP6.0 between 2030-2069.  Over the
SH ocean gyres that were brightened (Fig. 1), we see a very robust cooling, reaching 2 K at the
center of the South Pacific foamed region.  However, the cooling mixes rather well throughout
the SH.  Cross equatorial flow and changes in the Hadley Cell transmit this cooling into the NH





tropics through the mechanisms described in section 1.4, above. Some of this cooling in the NH
tropics is then transmitted to the NH extratropics.
G4Foam is significantly cooler ($p < 0.05$) than RCP6.0 in almost all locations south of
30°N, in mid latitude NH continental regions windward of the Atlantic and Pacific, and at very
high latitudes. Figure 3d shows that G4Foam is less effective in cooling extratropical NH land
regions during JJA. This is reasonable, since continental heating in the NH JJA season is more
dominated by local heating than the other seasons, in which meridional energy transport plays a
larger role. Figures 3a and 3c show that G4SSA is more effective over NH continents than
G4Foam. A key weakness of G4Foam, if implemented alone, would be its failure to adequately
reduce human suffering induced by heat stress in NH mid-latitudes during the summer as a result
of ongoing global warming.
Since the G4Foam forcing alone, with the amplitude of the current experiments, would be
insufficient to achieve any of the objectives of the G4Foam experiment, positive feedbacks that
enhance cooling and circulation responses must be triggered by the G4Foam forcing to enhance a
resulting cooler, wetter climate. Figure 4 shows change in low cloud fraction both year-round
and in the JJA season. The largest change is in the northern half of the regions where foam is
applied, and the area to the north of those foamed regions. The changes in low clouds in these
regions are both large and statistically significant. We attribute much of this low cloud response
to prevailing southeasterly winds in the SH tropics advecting cold air northeastward, through the
northern part of the foamed regions and then over the warmer tropical waters equatorward of the
foamed regions. This cold advection over relatively warm water forms broad regions of low
clouds in these subtropical and tropical regions (Fig. 4).
Another striking G4Foam feature is the large and statistically significant increase in low
clouds over land across central Africa, the Middle East and Southeast Asia. These low clouds
are coincident with the large cooling in Africa and the Middle East, particularly during the JJA
season relative to both G4SSA and RCP6.0 (Figs. 4c, 4d). These are very hot areas and heat
related mortality and morbidity are of great concern. A similar increase in low clouds is evident
in the tropical eastern Pacific. This is coincident with the mean northward displacement of the
ITCZ in G4Foam with respect to G4SSA and RCP6.0, not with any changes in the El Niño-
Southern Oscillation (ENSO).
In G4Foam, clouds are the key to changing the radiation budget in the tropics. In
G4Foam there is a change in shortwave cloud forcing of –2.32 W m$^{-2}$ annually and –2.59 W m$^{-2}$
during JJA, relative to G4SSA. Only very small increases in longwave cloud forcing of 0.42 W
m$^{-2}$ annually, and 0.07 W m$^{-2}$ in JJA counter this negative forcing. The overall change in cloud
radiative forcing in the tropics is –1.90 W m$^{-2}$ annually and –2.52 W m$^{-2}$ during JJA.
Total cloud fraction is shown in Fig. 5. Figs. 5c and 5d are particularly striking in
showing the increase in clouds over Africa and Southeast Asia during the JJA wet monsoon
season in those regions. Under G4Foam, these regions generally experience cloudier and cooler
summers relative to RCP6.0 and are cloudier and only very slightly warmer on average
compared to G4SSA. Some parts of the Sahel and the Middle East are actually slightly cooler in
G4Foam than RCP6.0. These changes in temperature and cloudiness play a key role in the
changes in the hydrological cycle under G4Foam, which we discuss next.
**3.2 Hydrological Cycle Response**
Relative to G4SSA, precipitation in G4Foam over land in the tropics increases by 3.9%
on an annual mean basis and by 4.9% during JJA (Table 1). Tropical precipitation in G4Foam
over land in the tropics increases by 1.4% on an annual mean basis and by 2.02% during JJA,





when compared to G4SSA. Each of these changes is statistically significant ($p < 0.05$).
Regarding the temperature change relative to G4SSA, G4Foam is only about 0.3 K warmer in
the tropics. The temperature dependence of precipitation, between 1.5% K$^{-1}$ and 3.0% K$^{-1}$,
(Emori and Brown, 2005), explains only a fraction of the precipitation increase. The statistically
significant increase in land-only precipitation in the tropics in G4Foam relative to RCP6.0 occurs
in a climate in which RCP6.0 is between 0.6 K and 0.7 K warmer than G4Foam, depending on
the season. Over the tropical oceans, in G4Foam, precipitation only increases by 0.8% on an
annual mean basis and 0.7% during JJA relative to G4SSA and there is a decrease of 1.6% on an
annual mean basis and a decrease of 1.9% during JJA relative to RCP6.0. These changes imply
that there is a large increase in moisture convergence over land in the tropics in G4Foam relative
to both G4SSA and RCP6.0.
Globally, over land, the response is similar, but the magnitude of change is a bit less.
Precipitation is statistically significantly increased over land in G4Foam relative to RCP6.0 by
about 0.5%. Precipitation is statistically significantly increased in G4Foam relative to G4SSA
over land by 3.5%.
The overall global precipitation difference between G4Foam and G4SSA or RCP6.0
when land and ocean are combined and all seasons and all latitudes are included are in line with
the magnitude of the temperature dependence of precipitation. Globally, G4Foam is warmer
than G4SSA by 0.3 K and there is 0.61% more precipitation. G4Foam is cooler than RCP6.0 by
0.6 K and drier by 1.98%.
The spatial pattern of precipitation changes is shown in Fig. 6. Precipitation is greatly
reduced over the ocean, particularly in the SH, relative to both G4SSA and RCP6.0. Changes in
precipitation poleward of 40° latitude in either hemisphere are largely due to the temperature
dependence of precipitation. The changes in the SH subtropics are dominated by the shortwave
forcing applied over the ocean gyres, which reduces both evaporation and precipitation in those
areas.
The changes in precipitation in the tropics are driven by the ITCZs northward shift.
Large precipitation anomalies occur in a narrow band north of the equator and smaller positive
anomalies occur in broader regions, primarily over NH monsoon regions. Importantly, we see a
statistically significant increase in monsoon precipitation over the Sahel, the Middle East, the
Indian subcontinent as well as southwest Asia and the maritime continent on an annual mean
basis in G4Foam relative to G4SSA (Figure 6a). Relative to RCP6.0, these changes are not
statistically significant over the Indian subcontinent or southwest Asia, but there are only very
isolated and small areas in these regions in which there is any precipitation reduction, either on
the annual mean or during JJA. Therefore, over much of heavily populated southern Asia, east
of the Arabian Sea, G4Foam will be cooler than RCP6.0 without any notable precipitation
differences.
Relative to both G4SSA and RCP6.0, there is a great deal more precipitation all year and
particularly during JJA over central America, the northern Amazon, much of Africa, parts of the
Arabian peninsula and the maritime continent. This response is more robust than the response
over Southeast Asia due to the more direct dependence of rainfall in these regions on ITCZ
position than in Southeast Asia, where the monsoon is also driven by numerous local and remote
factors, including ENSO and the Indian Ocean Dipole.
Although these G4Foam simulations are effective in enhancing moisture convergence
over heavily populated and highly cultivated regions, particularly in the tropics, there are regions
that would suffer under this regime. Precipitation patterns for islands in the South Pacific are





largely governed by the position and strength of the South Pacific Convergence Zone (SPCZ),
which changes substantially under G4Foam due in part to the cooling and to the movement of
gradients of temperature and pressure. Precipitation deficits over Madagascar and some regions
in Africa and South America exceed 10%. However, the very large precipitation deficits in
G4Foam are largely confined to the SH oceans and the land areas that tend to be far less
populated than in the areas where it will increase.

While the changes in precipitation are important and useful in describing the climate
response in G4Foam, the change in precipitation minus evaporation between G4Foam and
G4SSA or RCP6.0 is more relevant to how the changes in climate will actually impact people's
lives. Figure 7 shows precipitation minus evaporation. Specifically Fig. 7a shows that
precipitation minus evaporation in G4Foam is increased, and this increase is significant relative
to G4SSA, across the Sahel, all of South Asia, the Maritime Continent, Central America and the
northern Amazon. These are all heavily populated regions that are heavily cultivated. Figure 7b
shows a similar pattern, albeit with the regions significantly gaining water slightly suppressed in
coverage, when G4Foam is compared to the warmer RCP6.0 rather than G4SSA. Figures 7c and
7d show changes in water supply during JJA, the NH wet monsoon season, when water is likely
needed the most. Due to variability in the monsoon, there is more heterogeneity in the JJA
response than the annual response, particularly across Southeast Asia. The water gain, driven by
a combination of increased precipitation, lower temperature and increased cloudiness in these
heavily cultivated regions, is the most important benefit of G4Foam.

Figure 8 shows the differences of annual cycles from 2030-2069 for zonal mean
precipitation, zonal mean precipitation minus evaporation, and zonal mean precipitable water
between G4Foam and G4SSA and between G4Foam and RCP6.0. They illustrate the northward
displacement of the ITCZ, with positive precipitation anomalies progressing poleward as the
boreal summer monsoon progresses. Figure 8f shows the difference in the zonal mean annual
cycle for column integrated precipitable water between G4Foam and RCP6.0. The striking
feature here is that zonal mean precipitation is higher at key latitudes in the tropics, despite zonal
mean column integrated precipitable water being much lower at the same latitude. This further
illustrates the presence of increased moisture convergence further north, at latitudes with more
land and population, which explains the precipitation increase concurrent with the decrease in
available water vapor.

In Fig. 9, we quantify the impacts on agriculture by looking at the photosynthesis rate
anomalies between G4Foam and RCP6.0. There are small, but statistically significant increases,
in photosynthesis rate in G4Foam relative to RCP6.0 in much of Southeast Asia. The most
dramatic changes occur in Central America and parts of the northern Amazon, where the high
$CO_2$, relatively cool and very wet conditions promote agriculture.
**4 Discussion**

This paper is an analysis of a geoengineering climate model experiment. Although for
this experiment, global warming is reduced without negative impacts on precipitation, as was
found in previous stratospheric aerosol implementations, this does not argue for the
implementation of climate engineering. Any such decisions will need to balance all the risks and
benefits of such implementation, and compare them to those from other possible responses to
global warming.
**4.1 Summary**

G4Foam would reduce global mean surface temperature relative to RCP6.0 by 0.6 K for
the 40-year period starting 10 years after the implementation of geoengineering. Clear sky top of





atmosphere net shortwave flux is reduced by 1.5 W m$^{-2}$ in G4Foam relative to RCP6.0. This is
achieved primarily by the shortwave forcing over the subtropical SH ocean gyres. Before
accounting for feedbacks, temperature is more sensitive to the forcing applied in G4Foam than
G4SSA. However, global mean surface temperature in G4SSA 0.3 K lower than G4Foam
because of a larger change in clear sky top of atmosphere net shortwave flux of –4.0 W m$^{-2}$ (Fig.
2). Additionally, the latitudinal distribution of temperature reduction is different in G4Foam
than in G4SSA. G4SSA is most effective in cooling the NH continents, while G4Foam most
effectively cools the surface south of around 30°N (Fig. 3). Precipitation over land globally, in
the tropics, during JJA globally, and during JJA in the tropics is statistically significantly
increased in G4Foam relative to both G4SSA and RCP6.0 (Fig. 6). The combination of cooling
and increased precipitation over land in the tropics results in a statistically significant increase in
precipitation minus evaporation on an annual mean basis over Central America, the Northern
Amazon, the Sahel, the Indian Subcontinent, the Maritime Continent and Southeast Asia in
G4Foam relative to G4SSA (Fig. 7). All of these areas are very densely populated and heavily
cultivated. Water scarcity is a major issue in these areas and G4Foam describes a climate model
response in which there is global cooling but more water is made available to many people in
regions on the brink of severe water shortages. Both the changes in the spatial pattern and
magnitude of changes in temperature and precipitation are far too large to be explained by the
forcing alone. Instead, much of the temperature and hydrological response is the result of
powerful cloud feedbacks and changes in the tropical meridional overturning circulation induced
by the placement of the ocean albedo forcing.
**4.2 The hydrological response**
The dominant cause of the G4Foam hydrological response is the intensification of the
southern Hadley Cell and the northward migration of the ITCZ in response to the asymmetric
forcing. However, the precipitation response is not zonally homogeneous, as the regional and
local mechanisms are also important to the distribution of precipitation.
First, we address the increase in precipitation over Central America. For this, we turn to
literature concerning the decline of Mayan civilization in Central America. Summer insolation
in the NH began to decrease about 5,000 years ago. The ITCZ migrated southward. This
southward shift caused rainfall to decrease in the crucial summer growing season. Long
droughts and eventually water shortages contributed to the civilization's decline (Poore et al.,
2004). In G4Foam, the ITCZ moves northward and the areas in which Mayan civilization
flourished, including Belize, Guatemala and parts of Mexico, once again receive a great deal
more precipitation. This response is incredibly strong and consistent in each ensemble member
(Figs. 6-8).
The long mid-to-late 20[th] century Sahel drought was primarily caused by the ITCZ being
pushed southward by preferential cooling of the NH (Folland, 1986). In G4Foam, the reverse is
true. SH cooling pushes the ITCZ north, which generally explains the G4Foam precipitation
increase in the Sahel.
A surprising finding is that portions of the Arabian Peninsula equatorward of 20°S
experience precipitation increases of up to 1 mm day$^{-1}$ during the JJA season. However, this
northward migration of boreal summer precipitation is evident in the paleoclimate record.
Evidence of such precipitation is found in Fleitmann et al. (2003), who showed changes in $\delta^{18}$O
in cave stalagmites in Oman, which indicate increased rainfall in Oman under the influence of
northward movement of the ITCZ over the Indian Ocean in periods of relative warmth in the NH
relative to the SH.





Changes in precipitation over the Maritime Continent are partially attributable to large-
scale convergence and rising air in those regions, as they lie longitudinally between G4Foam
forcing zones where subsidence is enhanced. However, the Indian Ocean Dipole (IOD) (Cai et
al., 2012; Chowadry et al., 2012) and Subtropical Indian Ocean Dipole (SIOD) phenomena
discussed below are more likely the key drivers of the precipitation response over the Maritime
continent.
In its positive phase, the SIOD features anomalously warm SSTs in the southwestern
Indian Ocean, east and southeast of Madagascar, and cold anomalies of SST west of Australia.
Stronger winds prevail along the eastern edge of the SH subtropical high over the Indian Ocean,
which becomes intensified and shifted slightly to the south during positive SIOD events. This
results in more evaporation over the eastern Indian Ocean, which cools SSTs in the Indian Ocean
east of Australia (Suzuki et al., 2004). In the SIOD negative phase, the opposite is $_{true}$. There is
cooler water in the southwest Indian Ocean, near Madagascar and warmer waters to the east,
near Australia (Behera et al., 2001; Reason, 2001).
The negative phase of the SIOD features more precipitation in western Australia and the
Maritime Continent. This negative SIOD phase is consistent with the SST pattern in the Indian
Ocean forced by G4Foam. Therefore, the negative SIOD like mean state in G4Foam appears to
play a role in the enhanced rainfall in Northwestern Australia and the Maritime Continent.
Based on both local and global changes in circulation, we expected a very large increase
in the strength of the Indian Monsoon. In addition to the planetary scale changes associated with
the ITCZ and the Hadley cell, the position of the semi-permanent high in the subtropical
Southern Indian Ocean also plays a large role in modulating the Indian summer monsoon.
Negative SIOD events during boreal winter are often followed by strong Indian summer
monsoons. During a negative SIOD event, the subtropical high in the Indian Ocean shifts
northeastward as the season shifts from December, January, and February to JJA. This causes a
strengthening of the monsoon circulation, intensifying the Hadley Cell locally during the JJA
monsoon.
A negative IOD is associated with a weakened Asian monsoon and an increase in
precipitation over Australia and the Maritime Continent. In G4Foam, advection of cold water in
the Somali current into the equatorial western Indian Ocean creates a negative IOD-like response
that partially counters the combination of the global scale Hadley cell response and the forced
SIOD, dampening the overall increase in the Indian monsoon. This warm west, cold east mean
state in the equatorial Indian Ocean resembles a negative IOD mean state and it helps to explain
the enhanced precipitation response in the Maritime Continent and the lower than expected
increase in precipitation over the Indian subcontinent. The Asian monsoon and precipitation
over the Maritime Continent are also governed in part by ENSO. However, no changes in ENSO
were evident in G4Foam relative to G4SSA or RCP6.0. There is also no evident response of
ENSO amplitude or frequency to any of several different regimes of stratospheric
geoengineering (Gabriel and Robock, 2015).
**4.3 Caveats**
The technology does not presently exist to actually deploy a stable, highly reflective layer
of microbubbles on the actual ocean surface. While a stable, highly reflective, nondispersive
foam has been developed in a saltwater solution, appropriate for climate engineering, this foam
has not been tested outside the laboratory, much less on the surface of a large area of rarely
quiescent ocean. The foam has not been immersed in a medium in which bacteria are present,
and the interaction between the bacteria and the protein surfactant could damage the layer of





microbubbles. Also, even though the diameter of these microbubbles is on the order of $10^{-6}$ m,
the demand for surfactant would likely overwhelm our current production capacity of whatever
surfactant is chosen. The research on the engineering required to perform stratospheric
geoengineering by sulfate injection is much further along than research of microbubble
deployment, which is still in its earliest stages.
However, since development of microbubble technology is underway, it is worthwhile to
determine how such a technology could be applied in a manner that would address serious
climate issues. The progress being made in research associated with stratospheric
geoengineering actually enhances the relevance of researching the climate impact of this
particular ocean surface geoengineering approach as G4Foam was designed with an eye toward
concurrent deployment with stratospheric geoengineering in the event the stratospheric
geoengineering were to cause the precipitation deficits that many model studies have shown that
it might.
More fundamentally, the propriety of any attempt to impose a the G4Foam forcing in an
attempt to achieve the modeled G4Foam climate is premised on a value judgment that it is
desirable to develop a technology that could redistribute essential resources between nations in
an attempt to achieve a net benefit to humanity as a collective when it unknowingly creates a
local scarcity of these essential resources. To some extent, making this value judgment is
germane and is a prerequisite to the discussion of any form of geoengineering. Even though
G4Foam would be successful in increasing water supply in more heavily populated areas, water
supply will almost certainly be reduced in remote regions, such as South Pacific islands. Is it
ethical to pick winners and losers when the selection process is aimed at increasing the number
of winners and decreasing the number of losers? Hypothetically, if G4Foam worked as
described in this paper, from a purely consequentialist perspective, and with the sole objective
being increased utility for the human collective, G4Foam could be considered beneficial.
Finally, this paper is concerned with the climate response to surface albedo changes. We
do not examine how placing an actual layer of microbubbles in the ocean would change ocean
circulation or impact chemistry and biology in the ocean. Evaluating the changes in the ocean,
especially changes in its circulation that are caused by the surface albedo modification, is one of
the next issues to explore. The ocean regions we propose to brighten have low biological
productivity and weak currents, but the possibility of remote impacts, due to changes in
circulation having negative impacts on important ocean regions, is worth considering.
**4.4 Future research**
Whether or not a concurrent deployment of stratospheric geoengineering and ocean
albedo modification could cool the entire planet while maintaining or enhancing the hydrological
cycle, particularly in the tropics, is the next natural step in this research. Such research is
motivated by the need to determine whether some combination of geoengineering techniques can
be used to offset undesirable regional climate disparities that using one method of
geoengineering alone could induce.

**Acknowledgments.** This work is supported by U.S. National Science Foundation (NSF) grants
AGS-1157525, GEO-1240507, and AGS-1617844. Computer simulations were conducted on
the National Center for Atmospheric Research (NCAR) Yellowstone supercomputer. NCAR is
funded by NSF. The CESM project is supported by NSF and the Office of Science (BER) of the
U.S. Department of Energy.





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





**Table 1.** Changes in temperature and precipitation in G4Foam relative to both G4SSA and
RCP6.0, for the entire globe and for the Tropics (20°S-20°N) annually and in Northern
Hemisphere summer, for the 40-year period beginning 10 years after the start of climate
engineering.

| Global, 2030-2069 | G4Foam – G4SSA (% change) | G4Foam – RCP6.0 (% change) |
|---|---|---|
| Precipitation (mm/day) | +0.02 (+0.61) | -0.06 (-1.98) |
| Land precipitation (mm/day) | +0.07 (+3.19) | +0.01 (+0.32) |
| Ocean precipitation (mm/day) | −0.01 (−0.36) | −0.08 (−2.57) |
| Temperature (K) | +0.27 | −0.53 |
| Land temperature (K) | +0.63 | −0.44 |
| **Global, 2030-2069**, June-July-August | | |
| Precipitation (mm/day) | +0.02 (+0.70) | -0.05 (-1.85) |
| Land precipitation (mm/day) | +0.08 (+3.35) | +0.02 (+0.70) |
| Ocean precipitation (mm/day) | +0.01 (-0.29) | −0.08 (−2.51) |
| Temperature (K) | +0.32 | −0.60 |
| Land temperature (K) | +0.71 | −0.53 |
| **Tropical, 2030-2069** | | |
| Precipitation (mm/day) | +0.06 (+1.59) | −0.03 (−1.06) |
| Land precipitation (mm/day) | +0.16 (+3.93) | +0.07 (+1.43) |
| Ocean precipitation (mm/day) | +0.03 (+0.77) | −0.07 (−1.92) |
| Temperature (K) | +0.21 | −0.60 |
| Land temperature (K) | +0.43 | −0.61 |
| **Tropical, 2030-2069**, June-July-August | | |
| Precipitation (mm/day) | +0.06 (+1.52) | −0.03 (−0.84) |
| Land precipitation (mm/day) | +0.16 (+4.66) | +0.07 (+2.02) |
| Ocean precipitation (mm/day) | +0.03 (+0.67) | −0.06 (−1.61) |
| Temperature (K) | +0.18 | −0.61 |
| Land temperature (K) | +0.37 | −0.70 |






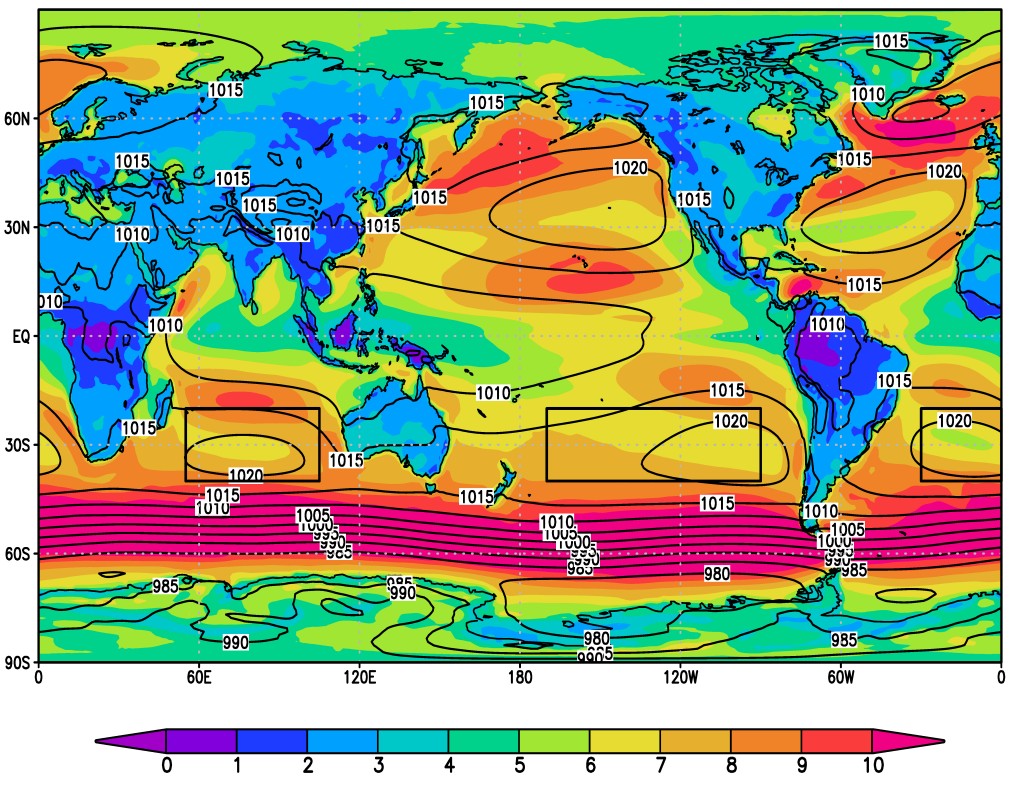

**Figure 1.** Applied forcing and global mean temperature response. Ocean albedo changed from
daily average of 0.0]6 to a fixed value of 0.15 over "foam regions," 20°N-20°S, 90°W-170°W
(South Pacific), 20°N-20°S, 30°W-0°E (South Atlantic) and 20°N-20°S, 55°E-105°E (South
Indian). Each "foamed" region is outlined in black. Control run sea level pressure (mb) is
shown with contours and 10-m winds (m/s) are shaded.





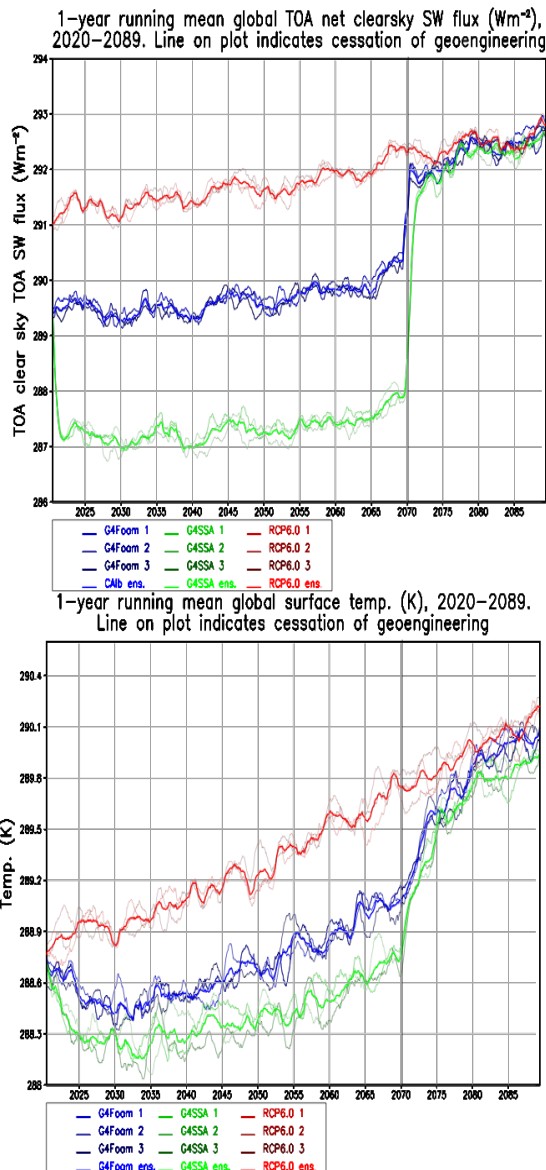

**Figure 2**. (a) Net clear sky SW flux at top of atmosphere, which includes the effects of changes
in radiation caused by changes in ocean surface albedo or land albedo (ice and snow), as well as
stratospheric aerosols (stratospheric geoengineering) and (b) Time series of global mean
temperature. In G4Foam, temperature is more than twice as sensitive to ocean albedo forcing as
it is to stratospheric geoengineering, as applied in G4SSA, albeit with very different latitudinal
distributions of temperature changes. Each ensemble member and the ensemble mean are shown
for each forcing.





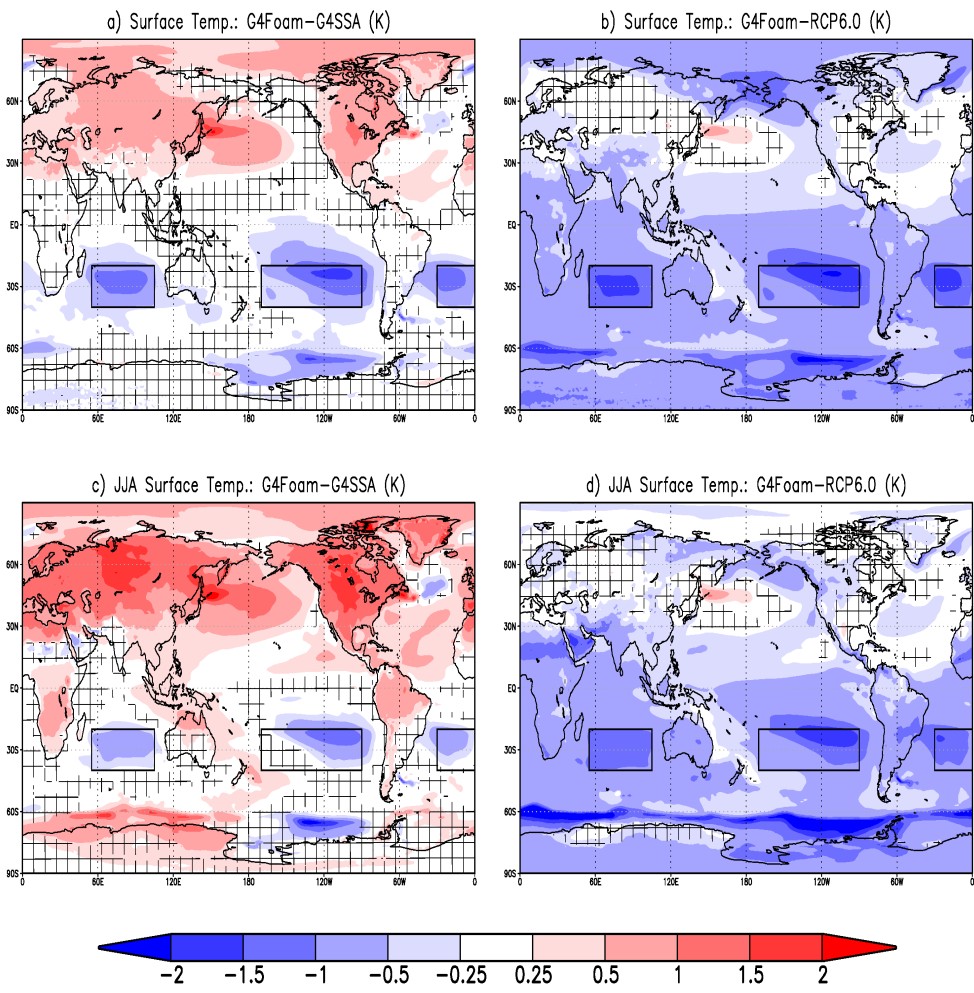

**Figure 3.** 2030-2069 surface temperature differences (K) between G4Foam and (a) G4SSA, (b)
RCP6.0, (c) G4SSA during JJA, and (d) RCP6.0 during JJA. Hatched regions are areas with $p >$
0.05 (where changes are not statistically significant based on a paired $t$-test). Black boxes
enclose foamed regions.




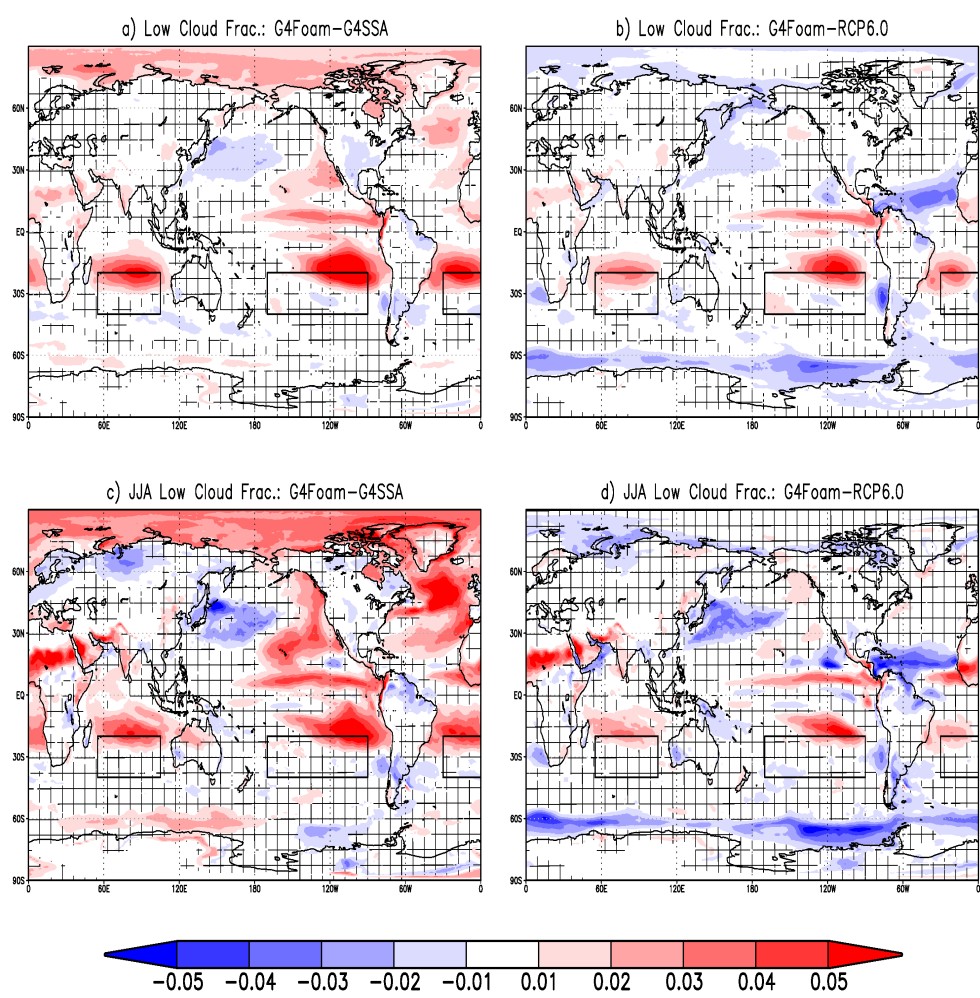

**Figure 4.** 2030-2069 low cloud fraction difference (unitless) between G4Foam and (a) G4SSA,
(b) RCP6.0, (c) G4SSA during JJA, and (d) RCP6.0 during JJA. Hatched regions are areas with
$p > 0.05$ (where changes are not statistically significant based on a paired $t$-test). Black boxes
enclose foamed regions.





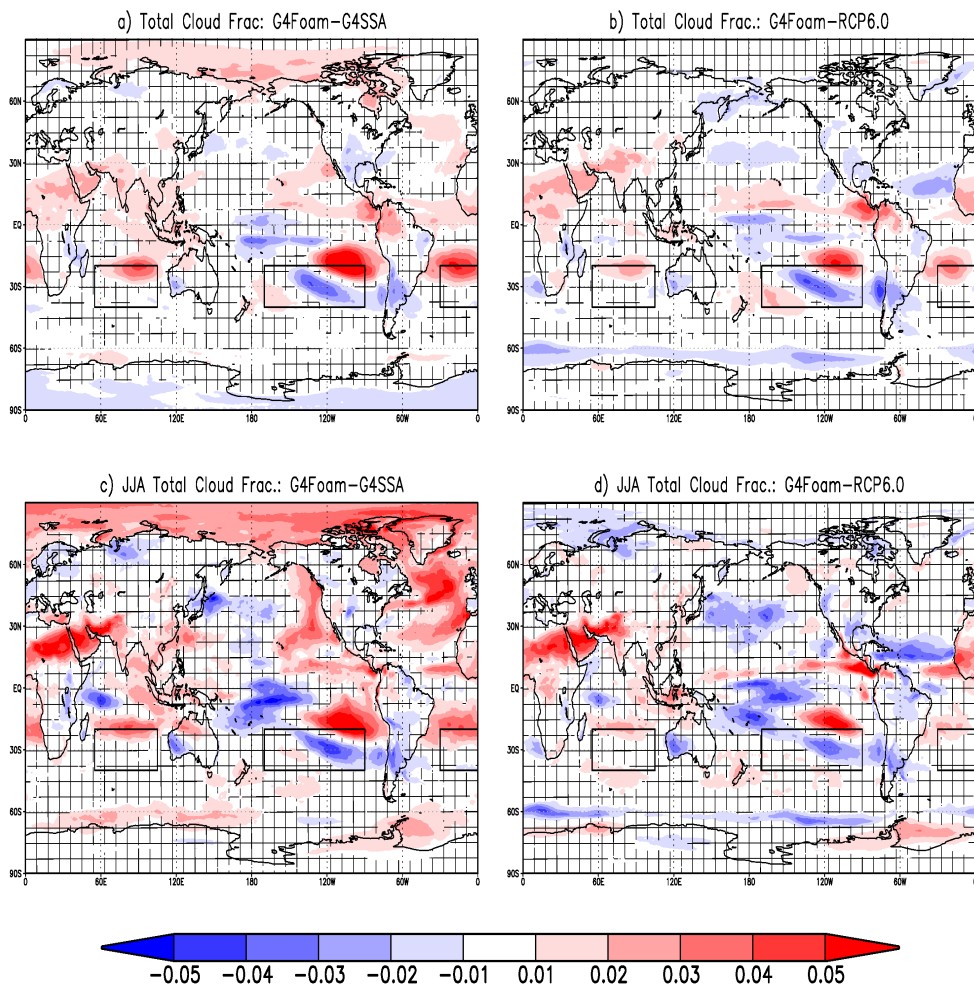

**Figure 5.** 2030-2069 total cloud fraction difference (unitless) between G4Foam and (a) G4SSA,
(b) RCP6.0, (c) G4SSA during JJA and (d) RCP6.0 during JJA.  Hatched regions are areas with
$p > 0.05$ (where changes are not statistically significant based on a paired $t$-test).  Black boxes
enclose foamed regions.





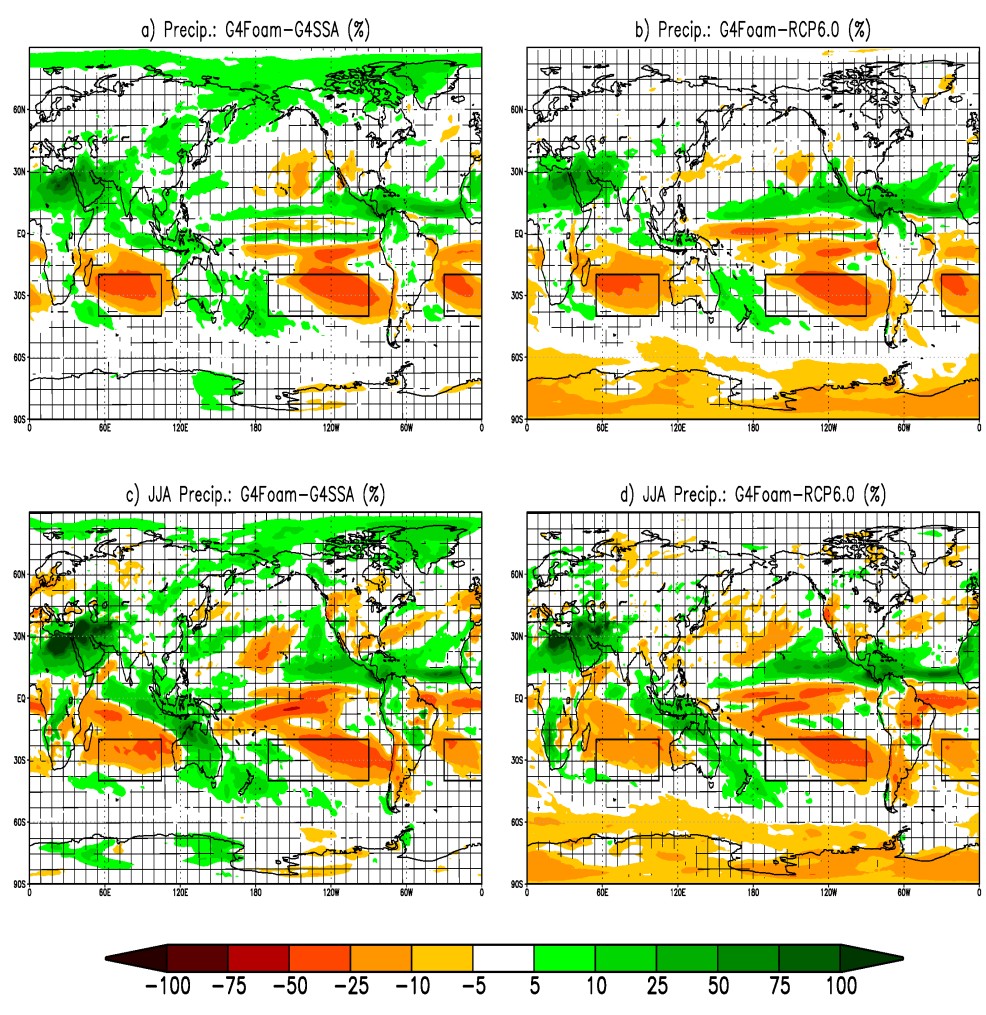

**Figure 6.** 2030-2069 precipitation difference (%) between G4Foam and (a) G4SSA, (b)
RCP6.0, (c) G4SSA during JJA and (d) RCP6.0 during JJA. Hatched regions are areas with $p >$
0.05 (where changes are not statistically significant based on a paired $t$-test). Black boxes
enclose foamed regions.



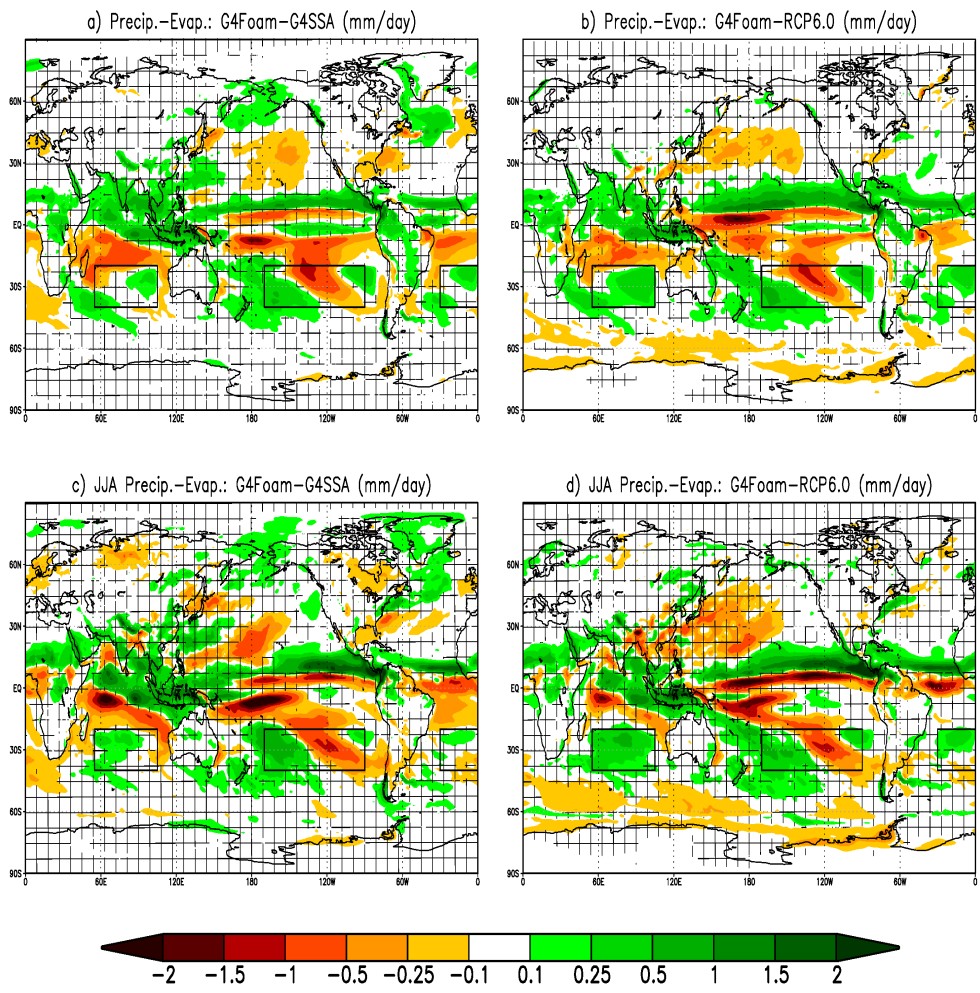

**Figure 7.** 2030-2069 precipitation minus evaporation difference (mm/day) between G4Foam
and (a) G4SSA, (b) RCP6.0, (c) G4SSA during JJA and (d) RCP6.0 during JJA. Hatched
regions are areas with $p > 0.05$ (where changes are not statistically significant based on a paired
$t$-test). Black boxes enclose foamed regions.





**Figure 8.** 2030-2069 monthly mean annual cycle of zonal mean precipitation (mm/day) for (a)
G4Foam minus G4SSA and (b) G4Foam minus RCP6.0, precipitation minus evaporation
(mm/day) for (c) G4Foam minus G4SSA and (d) G4Foam minus RCP6.0, and total precipitable
water (mm) for (e) G4Foam minus G4SSA and (f) G4Foam minus RCP6.0.





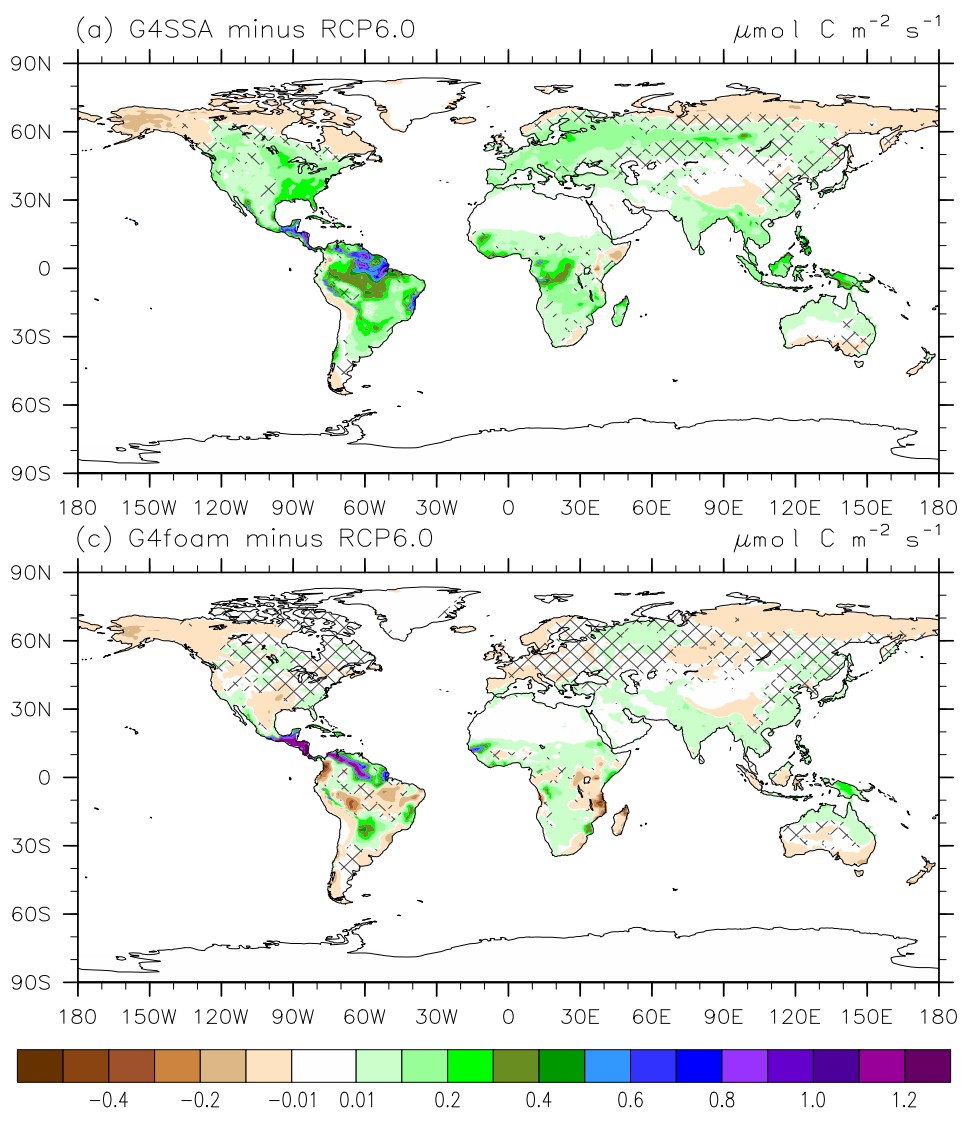

**Figure 9.** (a) Photosynthesis rate differences between G4SSA and RCP6.0 during years 2030–2069 (sulfate injection period, excluding the first 10 years) (Fig. 4a from Xia et al., 2016). (b) Photosynthesis rate anomaly between G4Foam and RCP6.0 during years 2030–2069 of solar reduction. Hatched regions are areas with p > 0.05 (where changes are not statistically significant based on a paired t test).