# Peer review of "The G4Foam Experiment: Global Climate Impacts of Regional Ocean Albedo Modification"

_Atmospheric Chemistry and Physics, 2016_

## Referee Comment (RC1) · Anonymous Referee #1 · 9 Nov 2016

**Title: The G4Foam Experiment: Global Climate Impacts of Regional Ocean Albedo Modification**

**General comments**

The paper deals with the G4Foam experiment, a unique testbed experiment for the GeoMIP simulations. G4Foam aims to increase the albedo of certain ocean regions. This method of climate engineering aims to reduce the global warming without reducing the monsoon rainfall. The study address the effect of G4Foam experiment on annual mean and JJA rainfall especially over land regions.

This paper addresses an interesting and socially relevant aspect of G4Foam experiment. Decrease in monsoon precipitation has always been a topic of discussion in climate engineering especially for G4SSA experiment. From the current study it is shown that the G4Foam experiment increases the precipitation over most of the land regions especially for JJA season with some adverse impacts elsewhere.

I recommend to accept the paper with the modifications/corrections suggested below.

**Specific comments**

1) In the case of termination of G4Foam, why not study a gradual termination instead of an abrupt termination? It would be interesting to see the recovering phase of G4Foam in a gradual termination process.

2) Paper claims the G4Foam experiment would cool the NH tropics and hence reduce the heat related mortality (Line no : 127, 377). However, heat related mortality is caused by extreme temperatures not the mean values, please justify.

3) Line no 382-386 : Please provide the values in comparison with RCP6.0?

4) From Table1 (also from Figure6) it is clear that the G4Foam experiment increases the tropical land precipitation by 1.4% annually and 2.02% during JJA relative to RCP6.0. How does this affect the extreme precipitation and frequent flooding events occurring in tropical land regions during monsoon time? Similar to reduction in precipitation, excess precipitation also affects the society right? So does this cause more adverse affects than benefits?

5) Line no : 461-463, (Similar to the above point) How can it be an important benefit without analysing the effects of increased precipitation especially during monsoon season. Extreme precipitation may lead to more floods adversely affecting the societies. Could you please justify this point with further analysis.

6) Line no : 472-474 Could you please give more explanation to the hypothesis.

7) Discussion part seems to be extremely positive about the precipitation response of G4Foam. Increase in precipitation does not always mean without negative impacts. Please rephrase the discussion with inclusion of the negative impacts of excess water supply and precipitation.

**Technical corrections**

Line no : 91   Please provide expansion of SSI.

Line no : 188  Could you please rephrase the sentence for better understanding.

Line no : 335  This is for JJA season right? please specify it.

Line no : 343  Is it G4Foam or G4SSA?

Line no : 396 Please check the value with the one given in Table1.

Line no : 398  Shouldn't it be RCP6.0 instead of G4SSA?

Line no : 406  Please check the values with Table1, values seems to be interchanged.

Line no : 856  Typo in Figure caption.

---

## Referee Comment (RC2) · Anonymous Referee #2 · 14 Nov 2016

This study examines the climate impacts of a unique geoengineering experiment that has ocean surfaces in certain regions brightened in a global climate model. The goal of the experiment, in which the ocean surface albedo is increased from 0.06 to 0.15 over three subtropical regions in the southern hemisphere, is to counteract the global warming without reducing monsoon rainfall. The results of this geoengineering experiment are compared to those of the specified stratospheric aerosols experiment and the RCP6.0 scenario to understand the response of temperature and precipitation. The results are interesting and the paper is generally well written. It fits the scope of ACP and in particular this geoengineering special issue. I have the following comments and suggestions for change, most of which are relatively minor.

1) The introduction session is a bit too long. Some of the background information for geoengineering in general, motivation and review can be shortened.

[Figure]

2) In many of the figures results are shown and discussed in terms of both annual mean and June-July-August (JJA) seasonal mean. It is unclear why JJA, which is neither austral summer nor the exact monsoon season in the northern hemisphere, is discussed in particular, as opposed to other seasons.

3) The color scheme in Figures 6-8 is different from that in Figures 3-5. This is fine, but using warm colors for decreases (i.e., negative changes) and cold colors for increases is a little inconvenient. Is there a particular reason for this?

4) Line 76 (also in the caption of Figure 1): the phrases of "daily average" and "fixed daytime value" are inconsistent and a little confusing. My understanding is that the albedo is changed from one constant value to another. Is that right?

5) Lines 88-99: please clarify the use of acronym SSI (versus SAI).

6) Lines 134: "the cloud feedbacks" are unclear.

7) Lines 248-249: Is this likelihood larger in this area than other areas in the SH? Please explain.

8) Lines 267-268: Is there a reference for the attribution of model improvements to finite-volume dynamical core?

9) Lines 310-311: is there a problem in the phrase inside the double quotes?

10) Line 339: needs some hyphens for "clear sky top of atmosphere"

11) Lines 341-346: it makes more sense to show net all-sky TOA flux in Fig. 2, maybe along with the net cloud forcing. The clear-sky forcing is not what is really exerted to the climate system.

12) Lines 366-373: need more evidence to support the explanation for the increase in low-cloud fraction over the three areas, where the relative humidity might have been already quite high. Why doesn't the increase occur in the entire downwind area?

[Figure]

13) Lines 418-421: Please elaborate on "the temperature dependence of precipitation".

---

## Author Comment (AC1) · 1 Dec 2016

Response to Referee 1 Comments on "The G4Foam Experiment: Global Impacts of Regional Ocean Albedo Modification," by C. J. Gabriel et al.

Referee comments are in black. Responses are in blue.

1) In the case of termination of G4Foam, why not study a gradual termination instead of an abrupt termination? It would be interesting to see the recovering phase of G4Foam in a gradual termination process.

It would be interesting to study a slower return to reference simulation conditions probably not only in G4Foam, but in other GeoMIP experiments as well. However, all GeoMIP experiments to date that have included termination, including G4SSA, have imposed abrupt termination. We keep to this convention to facilitate comparison with RCP6.0 and G4SSA. If one has simulated a large step response, it is straightforward to scale the results to a more gradual response. The abrupt change has the advantage of a large signal-to-noise ratio, so the response is easily identified.

2) Paper claims the G4Foam experiment would cool the NH tropics and hence reduce the heat related mortality (Line no : 127, 377). However, heat related mortality is caused by extreme temperatures not the mean values, please justify.

We have eliminated the assertion that G4Foam would reduce heat-related mortality, as we have found that this may not be true. The following text has been added to the manuscript. Please see lines 100-115 in the revised manuscript.

"The asymmetric cooling would force changes in the Hadley Cell, enhancing cross-equatorial flow, which would cool the surface in the NH tropics, especially during JJA, when heat mortality and morbidity is highest. However, despite a reduction in the JJA mean temperature in the tropics, extreme events are responsible for most heat-related mortality and morbidity, and the reduction in the mean temperature does not necessarily mean that there will be a reduction in the type of extreme heat events that cause human tragedy. While Kharin et al. (2007) showed that, in general, temperature extremes track with the mean temperature, this is not always the case. The changes in extreme events may, for example, be greater at high latitudes and the variability of temperatures over land may increase in a warmer climate.
"Specific to geoengineering, Aswathy et al. (2015) showed that different climate engineering methods produce spatially heterogeneous changes in extreme precipitation and temperature events. They showed that one SRM scheme may be more effective than another in reducing different types of extreme events despite relatively similar global and regional mean responses. In particular, a marine cloud brightening scheme that brightens ocean areas between 30°N and 30°S is shown to be less effective in reducing extreme precipitation and temperature events over land than the G3 experiment is."

Aswathy et al. (2015) used output from three different earth system models, each with multiple ensemble members, and performed detailed analysis of five variables related to extreme events. In the event more modeling groups run G4Foam, or we run

other similar test bed experiments, that type of analysis would be valuable.  Our goal in this testbed experiment is to describe the G4Foam experiment and describe some of the mechanisms that bring about the mean climate response.

3) Line no 382-386 : Please provide the values in comparison with RCP6.0?

We now also provide the values relative to RCP6.0.

4) From Table1 (also from Figure 6) it is clear that the G4Foam experiment increases the tropical land precipitation by 1.4% annually and 2.02% during JJA relative to RCP6.0. How does this affect the extreme precipitation and frequent flooding events occurring in tropical land regions during monsoon time? Similar to reduction in precipitation, excess precipitation also affects the society right? So does this cause more adverse affects than benefits?

This is an important point and we have removed references to the desirability or benefits of any of the respective hydrological regimes under G4SSA, G4Foam and RCP6.0 in the manuscript.  G4Foam was designed to cool Earth and increase precipitation, particularly in the tropics, relative to G4SSA. The fact that G4Foam produces this excess precipitation response relative to RCP6.0 is one of the reasons why we mention in 4.4 Future Research that we may combine stratospheric SRM with surface albedo modification to more effectively cool the planet without increasing precipitation to a level above that under RCP6.0 in already wet tropical areas. The manuscript has been adjusted.  We are endeavoring to portray a balanced picture of the climate effects of G4Foam.  We remain agnostic as to whether those climate effects are good or bad.  In particular, section 3.2 Hydrological Impacts now offers a balanced description of the results of G4Foam, as does 4.3 Caveats.  Future work that considers extreme events and natural resource economics may address whether the climate impacts brought about G4Foam ultimately can be rigorously characterized as more adverse or more beneficial both regionally and globally.

5) Line no : 461-463, (Similar to the above point) How can it be an important benefit without analyzing the effects of increased precipitation especially during monsoon season. Extreme precipitation may lead to more floods adversely affecting the societies. Could you please justify this point with further analysis.

We now emphasize that a precipitation increase in G4Foam relative to RCP6.0 is not the goal of G4Foam.  We have also withdrawn the claims about beneficial changes in water supply and instead only discuss changes in P-E.  More broadly, we have removed normative language about "benefits" and desirability" of the precipitation response, and instead just report the scientific results. This manuscript is designed to describe the results of the experiment and to describe the mean response and describe the relevant mechanisms.  To justify our points about G4Foam being beneficial to water supply, it would be necessary to study both extreme events and the economic, policy and resource allocation factors that determine the availability of water in a particular area.

6) Line no : 472-474 Could you please give more explanation to the hypothesis.

Lines 472-474 have been removed.  This was an oversimplification with little physical meaning.  The key here is the northward migration of the ITCZ and the global scale changes in the Hadley Cell.

7) Discussion part seems to be extremely positive about the precipitation response of G4Foam.  Increase in precipitation does not always mean without negative impacts. Please rephrase the discussion with inclusion of the negative impacts of excess water supply and precipitation.

We have revised the manuscript to portray a more balanced picture of the climate effects of G4Foam.  We remain agnostic as to whether those effects are good or bad. Specifically, we have added discussion to section 3.2 Hydrological Response to give more weight to both the negative effects of excessive rainfall in the tropics and the potential for adverse impacts due to reduced rainfall in the SH.  Section 4.3 Caveats also discusses potential problems with G4Foam.  Finally, the paper ends with section 4.4 Future Work.  While the climate response in G4Foam is robust in that it cools important regions and changes the spatial distribution of rainfall in a way that may be favorable for some, G4Foam has obvious deficiencies.  For example, NH land areas are not cooled very much, precipitation increases too much in already wet tropical regions, and parts of the SH receive a very large decrease in precipitation.  Additionally, since we do not aim to describe changes in the distribution of extreme events, we eliminate discussion of "water supply" and instead discuss precipitation minus evaporation.  A higher or lower amount of extreme precipitation events could increase or decrease runoff, which would then impact water supply independent of precipitation minus evaporation.

**Technical corrections**

Line no : 91 Please provide expansion of SSI.

Stratospheric sulfate injection (SSI) is now defined.

Line no : 188 Could you please rephrase the sentence for better understanding.

You are correct to point out that this sentence was confusing.  The purpose here was to describe the mechanism underlying the southward migration of the ITCZ.  We have clarified the sentence, which now reads "The forced cooling over the NH was enhanced by a positive dynamical feedback in the North Atlantic Ocean (Broccoli et al. 2006; Kang et al. 2008), and the ITCZ and associated tropical rainbelts migrated south." There is no need to bring up the energy-flux-equator here.

Line no : 335 This is for JJA season right? please specify it.

Yes.  During JJA added.

Line no : 343 Is it G4Foam or G4SSA?

We meant G4SSA and have changed G4Foam to G4SSA in that sentence.

Line no : 396 Please check the value with the one given in Table1.

The values in the table were correct.  We changed the text to reflect those values.

Line no : 398 Shouldn't it be RCP6.0 instead of G4SSA?

Yes.  We changed it to RCP6.0

Line no : 406 Please check the values with Table1, values seems to be interchanged.

We checked the values and there were a couple mistakes in the text.  We fixed those mistakes and the values in the text now match the values in Table 1.

Line no : 856 Typo in Figure caption.

Typo fixed.

*We have also shortened the abstract by one sentence.  Line 646-647 added to acknowledgements to thank you for your valuable comments.*

**References**

Aswathy, V. N., Boucher, O., Quaas, M., Niemeier, U., Muri, H., Mülmenstädt, J., and Quaas, J.: Climate extremes in multi-model simulations of stratospheric aerosol and marine cloud brightening climate engineering, Atmos. Chem. Phys., 15, 9593-9610, doi:10.5194/acp-15-9593-2015, 2015.

Kharin, V. V., Zwiers, F. W., Zhang, X., and Hegerl, G. C.: Changes in temperature and precipitation extremes in the IPCC ensemble of Global Coupled Model Simulations, J. Climate, 20, 1419– 1444, doi:10.1175/JCLI4066.1, 2007.

---

## Author Comment (AC2) · 1 Dec 2016

Response to Referee 2 Comments on "The G4Foam Experiment: Global Impacts of Regional Ocean Albedo Modification," by C. J. Gabriel et al.

Referee comments are in black. Responses are in blue.

1) The introduction session is a bit too long. Some of the background information for geoengineering in general, motivation and review can be shortened.

We agree and have removed the excess background information on geoengineering, reduced the length of the motivation section and the amount of literature review. Please see the new, ~35% shorter, introduction section. We were also able to remove some redundant language in sections 2-4 to make the paper a bit shorter.

2) In many of the figures results are shown and discussed in terms of both annual mean and June-July-August (JJA) seasonal mean. It is unclear why JJA, which is neither austral summer nor the exact monsoon season in the northern hemisphere, is discussed in particular, as opposed to other seasons.

JJA is chosen because it is meteorological summer in the NH and using JJA facilitates comparison with G4SSA, which reports results in terms of JJA (Xia et al., 2016). However, the Indian Monsoon season is typically defined JJAS, and we would use JJAS as our summer/wet monsoon season if we were focusing primarily on the Indian monsoon, or even exclusively on the Asian monsoon more broadly. Not all precipitation that is of interest in this study is monsoon precipitation, and various monsoon regions do experience somewhat different wet monsoon seasons. The cloud and temperature responses that are most of interest to highly cultivated and populated regions are best expressed by using JJA, since the NH is at its warmest during that meteorological season. Future work associated with the G4SSA and G4Foam simulation may look at, among other things, possible changes in monsoon onset and withdrawal in various geoengineering scenarios relative to what will happen under the RCP scenarios.
We add a summary of this reasoning to the text at lines 286-288 of the revised manuscript.

3) The color scheme in Figures 6-8 is different from that in Figures 3-5. This is fine, but using warm colors for decreases (i.e., negative changes) and cold colors for increases is a little inconvenient. Is there a particular reason for this?

Yes. The green is intended to signify a wet anomaly, and the brown is used to signify a dry anomaly. This color scheme is only used for hydrological variables precipitation, evaporation and precipitation minus evaporation (P-E). The colors we used are the traditional ones used for those variables, for example in the IPCC reports and in NOAA's Palmer Drought Index maps.

4) Line 76 (also in the caption of Figure 1): the phrases of "daily average" and "fixed daytime value" are inconsistent and a little confusing. My understanding is that the albedo is changed from one constant value to another. Is that right?

The albedo is actually changed from a value with a very small daily cycle that has a daily average value of 0.06 to a constant value of 0.15 (with no daily cycle) in the "foamed" regions. The inconsistent language has been removed. Please see that section, now at lines 50-55 and line 57-65, as well as the caption to Figure 1, which more clearly explains the change in albedo we imposed in the model. We have also added the caveat that an actual foamed region would likely exhibit fluctuations in albedo for many reasons and that additional study of the foam itself would be necessary to provide sufficient information to include fluctuations in foamed region albedo in future modeling studies. This could result in a slightly different surface energy budget than the constant albedo foam modeled here.

"RCP6.0 and G4SSA are run with an ocean surface albedo that contains a very small daily cycle, but the average albedo over a day is 0.06. The albedo of the ocean surface is raised from this daily mean of 0.06 to a constant value of 0.15, with no daily cycle, over the subtropical ocean gyres in the Southern Hemisphere, specifically 20°N-20°S, 90°W-170°W (South Pacific), 20°N-20°S, 30°W-0°E (South Atlantic) and 20°N-20°S, 55°E-105°E (South Indian) (Fig. 1). Everywhere else, ocean surface albedo in G4Foam is calculated the same as in RCP6.0 and G4SSA."

5) Lines 88-99: please clarify the use of acronym SSI (versus SAI).

This was an error in editing. We now define stratospheric sulfate injections as SSI in the revised manuscript and SSI is used exclusively throughout to refer to stratospheric SRM. There is no mention of "SAI" any longer.

6) Lines 134: "the cloud feedbacks" are unclear.

We have changed "the cloud feedbacks" to "any cloud feedbacks." We are acknowledging that the effectiveness of the G4Foam forcing will be affected by how clouds respond to the forcing, that the nature of this response is unknown until we conduct the experiment, and that we consider clouds to potentially be a large source of uncertainty. Please see lines 122-125.

7) Lines 248-249: Is this likelihood larger in this area than other areas in the SH? Please explain.

You are correct to point this out. The likelihood is not necessarily larger and the reference to that likelihood has been removed. We were principally motivated to brighten those specific regions because of their low cloud fraction, low wind speeds, weak currents, and lack of biological productivity.

8) Lines 267-268: Is there a reference for the attribution of model improvements to finite-volume dynamical core?

Yes. The reference to Neale et al. (2013) has been added at line 228.

9) Lines 310-311: is there a problem in the phrase inside the double quotes?

No.  We have removed the quotes.

10) Line 339: needs some hyphens for "clear sky top of atmosphere"

Hyphens added.

11) Lines 341-346: it makes more sense to show net all-sky TOA flux in Fig. 2, maybe along with the net cloud forcing. The clear-sky forcing is not what is really exerted to the climate system.

We agree and have added the new Figure 2, which shows net all-sky TOA flux (Figure 2a) and net cloud forcing (Figure 2b).The beginning of section 3.1, now at lines 290-305, now refers to the new Figure 2. Additionally, we now report changes in radiative forcing as the all-sky values, rather than the clear-sky values, since all-sky is what is actually exerted on the climate.  Figure 3, showing clear-sky forcing, which is very similar to, and at the beginning of the simulation, is almost exactly equal to, the imposed ocean surface albedo forcing.  Clear-sky SW TOA is now only shown to illustrate that the G4Foam forcing is more efficient in achieving cooling than G4SSA forcing.

12) Lines 366-373: need more evidence to support the explanation for the increase in low-cloud fraction over the three areas, where the relative humidity might have been already quite high. Why doesn't the increase occur in the entire downwind area?

We have revised the manuscript to provide a detailed explanation for the increase in low cloud fraction in the areas to the north and northeast of the three "foamed" regions.  The new section is copied below and can be found at lines 329-373:

"The low cloud fraction increase in the three areas to the north and northeast of the G4Foam-forced subtropical surface regions is likely due to a stronger than normal trade wind inversion (TWI).  The inversion develops when warm air is trapped above the atmospheric mixed layer due to large-scale subsidence and surface mixing of cooler air above these relatively low SST regions.  The increase in low cloud fraction does not occur over the entire downwind area because SSTs increase from east to west, causing a change in the lower troposphere as you travel from east to west.  Moving west, the stratocumulus layer, which is trapped under the inversion base, decouples from the mixed layer in the lower troposphere.  The surface warming triggers more turbulence within the planetary boundary layer, which allows for enhanced cumulus mixing in the cloud layer, which entrains dry air, and the marine stratocumulus layer evaporates as you travel west.
"The subtropical high-pressure systems are stronger in G4Foam, due to the stronger than normal Hadley Cell, which enhances subsidence throughout the subtropics.  Typically, a subsidence inversion is strongest over the center of the subtropical anticyclones, over cold currents (particularly the Peru Current), and over cooler than normal waters, which are subjected to enhanced upwelling in large part by trade winds on the periphery of the subtropical highs (DeSzoeke et al., 2016).  The TWI becomes weaker and its base increases in height with distance

towards the west and towards the equator as SSTs increase. This pattern is particularly evident in the Pacific, due to the larger geographical extent of the forced area.

"Specifically, under G4Foam conditions, the increased low cloud fraction areas are the result of the combination of enhanced large-scale subsidence (stronger Hadley cell) and a cooler than normal ocean surface. The cooler than normal surface waters are due to general cooling throughout the SH, as well as an increase in wind-driven upwelling over these areas of increased low cloud fraction, which are already prone to upwelling, large fraction of low clouds and high relative humidity.

"In these areas north of the foamed areas, the subsidence inversion is not quite as strong as it is right under the subtropical high. However, SSTs are artificially low, due to general cooling of the hemisphere and enhanced upwelling, driven by anomalously strong winds, and mixing of this anomalously cool surface air within the planetary boundary layer keeps the lowest levels of the atmosphere cool, keeping the marine air inversion base above the lifting condensation level, allowing stratocumulus clouds to form at low altitude, below the base of the inversion. Additionally, since SST is lower than air temperature in the areas of enhanced low clouds, the surface inversion is further maintained as a result of sensible heat flux from the atmosphere to the ocean. Ultimately, the strong inversion often results in more marine layer cloud formation and longer times for the clouds to dissipate. This response is consistent through the 2030-2069 period. This enhanced low-cloud fraction response is similar to the seasonal cycle of marine low clouds around the periphery of the subtropical highs (Wood and Bretherton, 2004; Chiang and Bitz, 2005; Wood and Bretherton, 2006; George and Wood, 2010; Mechoso et al., 2014).

"The relationship between the strength of the subtropical high, inversion strength and marine cloud prevalence can be elucidated by analogy to the behavior of the very well-observed marine low clouds off of the California coast. The strength of the inversion, and the prevalence of marine low clouds are modulated by the annual cycle with annual maximum low cloud extent in the summer, when the subtropical high is at its strongest.

"The increased low cloud fraction response is not seen above the actual G4Foam forced regions despite the cooler SST. The subsidence is so strong in these areas that the base of the inversion falls below the lifting condensation level, and few clouds form."

13) Lines 418-421: Please elaborate on "the temperature dependence of precipitation".

We have clarified this portion of section 3.2. It is rather evident that with global warming, specific humidity in the tropical planetary boundary layer will increase by 7% $K^{-1}$, scaling with Clausius-Clapeyron (e.g., Held and Soden, 2006). However, the processes involving precipitation are quite complex and while it is clear that global mean precipitation will increase as global mean temperature increases, there is a wide range of estimates in the literature of how much precipitation will increase per degree of global warming. In the revised manuscript, we refer to a review that collects estimates from the literature of how much precipitation will increase per degree of global warming. They estimate a 1.5%-3% $K^{-1}$ range.

We then report the precipitation change in G4Foam, relative to both G4SSA and RCP6.0 and note that while global mean precipitation over land and ocean changes by about 2%-3% per degree of global mean temperature, the changes over land, especially over the tropics, are dramatically different. Precipitation actually increases over land in G4Foam relative to RCP6.0, despite 0.6 K of cooling and there is far more precipitation over land in G4Foam than G4SSA

despite G4Foam being only slightly warmer.  We've clarified the discussion in the revised manuscript.

*We have also shortened the abstract by one sentence.  Line 646-647 added to acknowledgements to thank you for your valuable comments.*

**References**

Held, I. M. and Soden, B. J.: Robust responses of the hydrological cycle to global warming, J. Climate, 19, 5686–5699, 2006.

Neale, R., Richter, J., Park, S., Lauritzen, P., Vavrus, S., Rasch, P. and Zhang, M.: The mean climate of the Community Atmosphere Model (CAM4) in forced SST and fully coupled experiments, J. Climate, 26, 5150–5168, 2013.

Xia, L., Robock, A., Tilmes, S., and Neely III, R. R.: Stratospheric sulfate geoengineering could enhance the terrestrial photosynthesis rate, Atmos. Chem. Phys., 16, 1479-1489, doi:10.5194/acp-16-1479-2016, 2016.